# Inhibition of the nucleolar RNA exosome facilitates adaptation to starvation

**Xi Feng[1], Xiaoman Wang[1], Shouhong Guang[2], Shanshan Pang[1]\*, Haiqing Tang[1]\***

1 School of Life Sciences, Chongqing University, Chongqing, China, 2 Department of Obstetrics and Gynecology, The First Affiliated Hospital of USTC, The USTC RNA Institute, Ministry of Education Key Laboratory for Membraneless Organelles & Cellular Dynamics, School of Life Sciences, Division of Life Sciences and Medicine, Biomedical Sciences and Health Laboratory of Anhui Province, University of Science and Technology of China, Hefei, China

\* sspang@cqu.edu.cn (SP); hqtang@cqu.edu.cn (HT)

## Abstract

In response to nutrient scarcity, cells must reallocate their limited energy for cellular maintenance at the expense of certain processes. How such a tradeoff is achieved remains largely unknown. RNA surveillance is crucial for the integrity of the transcriptome, whose defects are associated with several human diseases. Unexpectedly, we discover that the nucleolar RNA exosome, a key RNA surveillance machine, is inhibited by starvation. This is not merely the cessation of a temporarily non-essential process, but rather a key signal to allocate energy. By rewiring one-carbon metabolism, the inhibition of RNA exosome reduces translation, the most energy-consuming process. Energy is then conserved for fat synthesis to enhance cellular maintenance and starvation survival. Notably, while benefiting starvation fitness, RNA exosome inhibition impairs the life span of well-fed animals, indicating a tradeoff between short-term and long-term fitness. Our findings suggest that the nucleolar RNA surveillance can be temporarily sacrificed to facilitate starvation adaptation.

## Introduction

Food scarcity is a significant challenge for organisms in their natural habitats. Organisms have evolved sophisticated mechanisms to cope with starvation and to maximize their survival. Among the adaptive responses, reprogramming of nutrient metabolism is crucial for survival during fluctuations in food supply. For example, as the primary energy reserve, the stored fat is degraded through lipolysis to liberate fatty acids (FAs), which are then catabolized by FA oxidation to generate acetyl-CoA for energy production. This adaptation ensures energy supply when external nutrients are limited. Over the past decades, how metabolic enzymes are regulated in response to starvation has been intensively studied in multiple organisms, and an elaborate and conserved gene regulatory network has been established [1–7].

**Data availability statement:** The RNA sequencing data have been deposited in the Sequence Read Archive (SRA) with an accession number of SRP471409. All other data are within the paper and its Supporting information files.

**Funding:** This work was supported by the National Natural Science Foundation of China (grant no. 32370828 and 32070754 [to H. T.], and grant no. 32271212 and 32071163 [to S. P.]), Natural Science Foundation of Chongqing, China (grant no. cstc2021ycjh-bgzxm0138 [to S. P.]), and the Fundamental Research Funds for the Central Universities (grant no. 2024CDJXY016). The funders had no role in study design, data collection and analysis, decision to publish, or preparation of the manuscript.

**Competing interests:** The authors have declared that no competing interests exist.

**Abbreviations :** cRT-PCR, circular reverse transcription PCR; DMSO, dimethyl sulfoxide; DR, dietary restriction; DTT, dithiothreitol; ER, endoplasmic reticulum; FAs, fatty acids; MA, malic acid; OA, oleic acid; PC, phosphatidylcholine; SAMe, S-adenosylmethionine; UPR, unfolded protein response; WT, wild-type; 1CM, one-carbon metabolism; 10-fTHF, 10-formyl-THF.

Fat metabolism is not an isolated process within cells but rather is intricately interconnected with many other cellular processes [8–10]. How is fat metabolism controlled in the complex subcellular environment? In other words, cells must coordinate entire cellular processes to meet the rewiring of fat metabolism, rather than simply exerting direct control over fat metabolic enzymes. This fundamental concept remains poorly explored. While certain cellular processes are vital for normal physiological function in well-nourished animals, they may be considered non-essential during fasting due to their energy-consuming characteristics. Therefore, these non-essential processes may be temporarily halted in order to conserve energy, which can then be utilized for lipid synthesis or directly by survival-critical processes. It remains largely unknown which cellular processes are temporarily "non-essential" and how they contribute to fat reserve maintenance and starvation survival.

Stress resistance plays a crucial role in defending against multiple environmental and cellular insults and has been regarded as a crucial determinant of cell preservation and survival [11]. Intriguingly, nutrient deprivation can cause the accumulation of stresses, such as ER stress and oxidative stress, in multiple species [12–14]. Moreover, the survival of starved organisms correlates with their resistance to stresses such as oxidative stress [15], suggesting that stress responses may belong to essential processes that need to be maintained during food scarcity. Notably, research on model organisms indicates that lipid species, like various FAs, are necessary for stress resistance [16]. This suggests that starvation survival may depend on the maintenance of stress responses through rewired fat metabolism.

The integrity of the cellular transcriptome is maintained by RNA surveillance machines that detect and degrade aberrant RNAs. The nuclear RNA exosome, a major RNA surveillance machine, is a conserved 3′ to 5′ exonucleolytic complex that processes and degrades both aberrant noncoding RNAs (e.g., rRNA) and protein-coding mRNAs, thereby playing crucial roles in diverse cellular processes, such as differentiation and development [17–20]. Accordingly, genetic mutations in RNA exosome-related genes cause various human disorders [21,22]. In this study, using *Caenorhabditis elegans* as a model organism, we unexpectedly found that RNA exosome perturbation can be remarkably advantageous for cellular maintenance, particularly in the context of nutrient deprivation. Starvation depletes the RNA exosome from the nucleoli, which acts as a signal to direct energy reallocation. During this process, the one-carbon metabolism (1CM) is reprogrammed to suppress cellular translation, one of the most energy-intensive processes; thus, the energy is conserved as fat for stress resistance and survival. As nucleolar exosome activity is crucial for rRNA quality control, we propose that cells interpret RNA exosome inactivation as a "no more translation" signal, thereby determining the cellular tradeoff in response to nutrient scarcity.

## Results

### Inactivation of the RNA exosome enhances systemic stress resistance

RNA surveillance mediated by the nuclear RNA exosome is essential for cellular function. During a genetic screening for new factors governing stress response in *C. elegans*, we unexpectedly discovered that RNAi inactivation of *exos-8*, a component

gene of the RNA exosome, activated the oxidative stress response reporter, *gst-4p*::GFP (Fig 1A). The effects were not limited to *exos-8*, as RNAi of other subunits of the RNA exosome (Fig 1B) also induced *gst-4p*::GFP expression (S1A Fig). More strikingly, the effects of the RNA exosome on stress response appear to be general, as RNAi of either *exos-8* or *exos-4.2* induced the expression of several other stress response reporters, including MTL-1::GFP, *hsp-4p*::GFP, *hsp-6p*::GFP, and *hsp-16.2p*::GFP (S1B–S1E Fig), which indicate DAF-16-dependent stress response, endoplasmic reticulum (ER) unfolded protein response (UPR), mitochondrial UPR, and heat stress response, respectively. qPCR analysis confirmed that the endogenous expression of most of these genes, except *hsp-4*, was increased by RNAi targeting *exos-8* or *exos-4.2* (S1F Fig). These data suggest that RNA exosome inactivation triggers a systemic activation of stress responses.

These findings were surprising because RNA exosome defects are generally believed to be detrimental for cells and organisms. Thus, we asked whether RNA exosome inactivation can indeed improve animal resistance to multiple stresses. We found that RNAi of *exos-8* or *exos-4.2* enhanced animal survival in response to oxidative stress (tert-butyl hydroperoxide), mitochondrial stress (malic acid, MA), ER stress (dithiothreitol, DTT), and heat stress (35 °C) (Fig 1C–1F). Additionally, mutation of *exos-10*, a catalytic subunit of the RNA exosome, also enhanced resistance to multiple stresses (Fig 1G–1J). We propose that since RNA exosome inactivation is interpreted as harmful, cells respond by inducing systemic cellular maintenance, which is critical for cell survival.

## Inhibition of the RNA exosome by nutrient deprivation

The nucleolar RNA exosome plays a crucial role in rRNA surveillance. The nucleolus has recently emerged as a cellular stress center [23,24]. Consistent with this, nucleolar expression of the RNA exosome is affected by temperature stress in *C. elegans* [25]. EXOS-1 is a core subunit of the RNA exosome that predominantly localizes and functions in the nucleolus. Mutations of the other exosome subunits deplete GFP::EXOS-1 from the nucleoli, which might indicate RNA exosome inactivation [26]. We verified that *exos-8* RNAi induced the depletion of GFP::EXOS-1 in the nucleoli (S2A Fig). Then, we investigated nutrient deprivation, the survival during which correlates with stress responses, and found that GFP::EXOS-1 was depleted from the nucleoli in response to starvation (Fig 2A). Interestingly, oxidative stress, mitochondrial stress, and ER stress also depleted GFP::EXOS-1 from the nucleoli (S2B–S2D Fig). As stress may inhibit food intake and cause a starvation-like state, we measured the pharyngeal pumping rate of worms and observed that these stresses all have profound suppressive effects on food uptake (S2E Fig), suggesting that GFP::EXOS-1 may respond to stresses in a way dependent on food intake. Although we do not have a definitive answer to this question, the above data led us to focus on nutrient deprivation in the current study.

To corroborate the regulation of the RNA exosome by starvation, we generated an EXOS-8::mCHERRY reporter strain and observed that EXOS-8 was also depleted from the nucleoli in response to starvation (Fig 2B). Additionally, starvation also depleted the nucleolar expression of GFP::EXOS-10 and mCHERRY::DIS-3 (Fig 2C and 2D), the catalytic subunits of the RNA exosome. These results suggest that starvation depletes the nucleolar expression of the RNA exosome.

To examine the activity of the RNA exosome during starvation, we investigated rRNA processing, a process governed by the nucleolar RNA exosome [18,20]. Using circular RT-PCR (cRT-PCR) to analyze rRNA intermediates [27], we first assessed the role of the *C. elegans* RNA exosome in rRNA processing by studying *exos-10* mutants, which lack the catalytic subunit of the RNA exosome. Our results showed that *exos-10* mutants accumulated two 18S intermediates (Fig 2E and 2F) compared to wild-type (WT) controls, indicating that the RNA exosome is essential for processing these pre-rRNAs. Next, we analyzed rRNA processing under starvation conditions, hypothesizing that if starvation suppresses RNA exosome activity, it would lead to the accumulation of the same pre-rRNAs observed in *exos-10* mutants. Consistent with this hypothesis, starved animals showed significant accumulation of these 18S pre-rRNAs (Fig 2G). These findings suggest that starvation inhibits the nucleolar RNA exosome, likely through the depletion of its nucleolar expression.

mTOR/TOR is a key sensor of nutrient scarcity, which is suppressed to promote adaptive response during starvation [28,29]. Using TOR pathway-related mutants, we next examined whether TOR is involved in exosome regulation. We

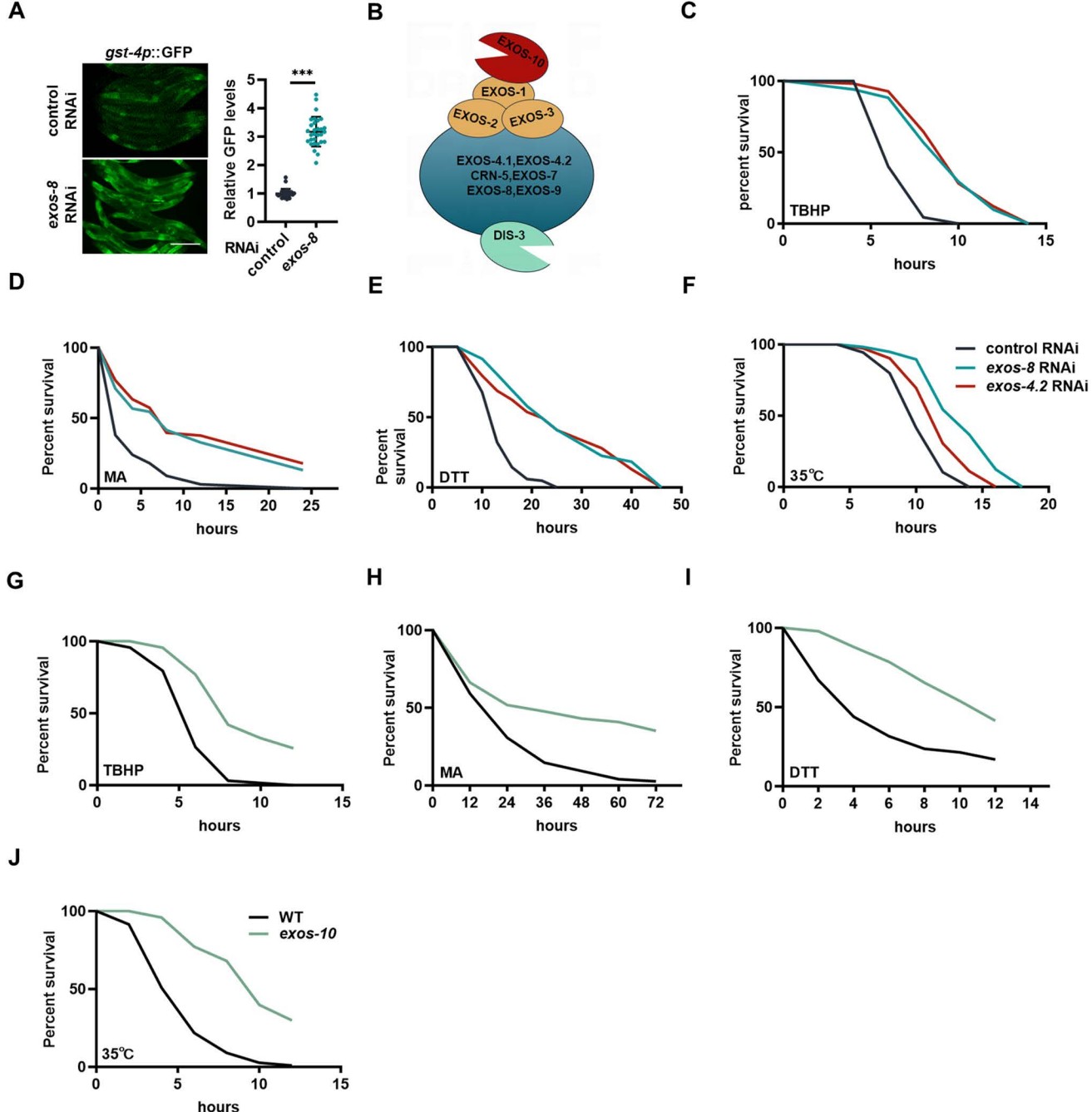

**Fig 1. Inactivation of the RNA exosome enhances systemic stress resistance. (A)** Left: Effect of *exos-8* RNAi on the expression of *gst-4p*::GFP, an oxidative stress response reporter construct containing GFP driven by the endogenous promoter of *gst-4*. Scale bar = 200 μm. Right: Relative GFP intensity. Unpaired two-tailed *t* test (*n* = 30 worms). **(B)** Schematics of the RNA exosome in *Caenorhabditis elegans*. The RNA exosome comprises the core complex and two catalytic subunits, EXOS-10 and DIS-3. The core complex contains six RNase pleckstrin homology (PH)-like proteins (blue) and three S1/K homology (KH) proteins (yellow). **(C–F)** Effects of *exos-8* and *exos-4.2* RNAi on oxidative stress resistance (C), mitochondrial stress resistance (D), endoplasmic reticulum stress resistance (E), and heat stress resistance (F). **(G–J)** Effects of *exos-10 mutation* on oxidative stress resistance (G), mitochondrial stress resistance (H), endoplasmic reticulum stress resistance (I), and heat stress resistance (J). Data are presented as mean ± SD. ***p < 0.001. S1 Table provides all repeats and statistical analyses of the survival experiments, where Repeat 1 of each experiment was used for generating the graphs. The numerical data presented in this figure can be found in S1 Data.

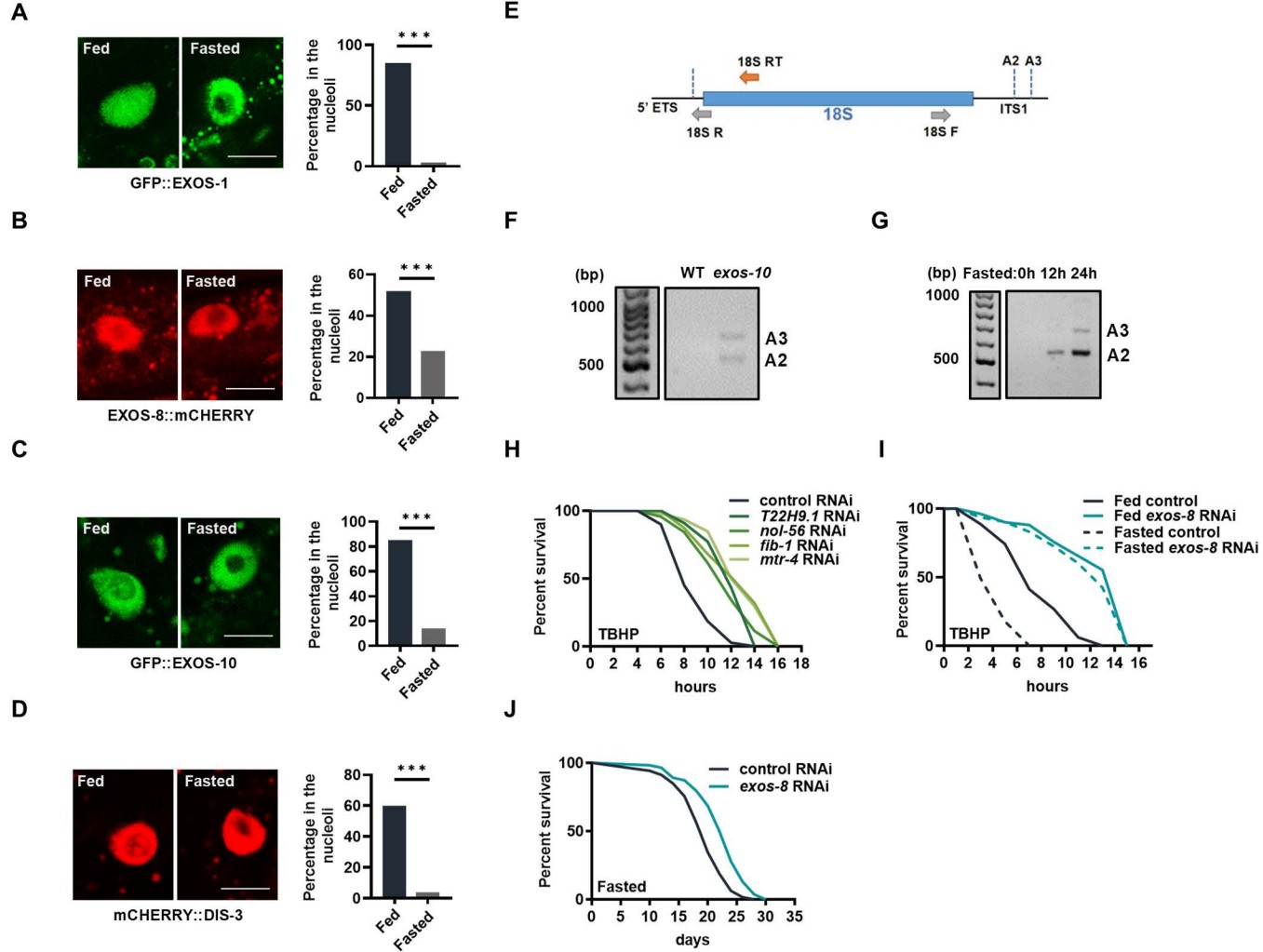

**Fig 2. Starvation depletes the nucleolar RNA exosome. (A–D)** Left: Effects of 6-h starvation on the nucleolar localization of GFP::EXOS-1 (A), EXOS-8::mCHERRY (B), GFP::EXOS-10 (C), and mCHERRY::DIS-3 (D) in the intestinal cells of day 1 adult worms. Scale bar = 10 µm. Right: Percentage of fluorescent signals in the nucleoli. $n = 77–145$ cells. **(E)** Schematics of the 18S rRNA and its intermediates. 18S RT represents the reverse transcription primer used for cRT-PCR, while 18S F/R are the PCR primers used for cRT-PCR. The dotted lines mark the end positions of the pre-rRNAs, with A2 and A3 indicating the 3′ end positions of two distinct pre-rRNAs. ITS refers to the internal transcribed spacer. **(F, G)** Effects of *exos-10* mutation (F) and starvation (G) (12 or 24 hours in L4 stage worms) on the levels of 18S pre-rRNAs. **(H)** Oxidative stress resistance in animals treated with RNAi targeting nucleolar genes that affect the nucleolar occupancy of the RNA exosome. **(I)** Effects of *exos-8* RNAi on oxidative stress resistance in well-fed and 12-h ur starved day 1 stage nematodes. **(J)** Effects of *exos-8* RNAi on the life span of starved animal. Data are presented as mean ± SD. ***$p < 0.001$. S1 and S2 Tables provide all repeats and statistical analyses of the survival experiments, where Repeat 1 of each experiment was used for generating the graphs. The numerical data presented in this figure can be found in S1 Data.

found that the RNA exosome subunits were precluded from the nucleoli by both the *rsks-1/S6k* and *rict-1/Rictor* mutations (S2F–S2H Fig), similar to starvation, suggesting that nutrient deficiency inactivates the nucleolar RNA exosome in a way that may partially depend on TOR.

## Inactivation of the RNA exosome benefits fitness in response to nutrient scarcity

Why does starvation prohibit the nucleolar RNA exosome? We hypothesized that nutrient deficiency is detrimental to stress resistance and that the purpose of nucleolar RNA exosome inhibition is to maintain stress resistance and,

ultimately, survival. While it is currently difficult to manipulate the RNA exosome specifically in the nucleolus, a previous study identified several nucleolar genes (*T22H9.1, snu-13, nol-56, fib-1, and mtr-4*) that control the nucleolar occupancy of the exosome subunit EXOS-10 [26]. We verified that RNAi targeting these genes depleted GFP::EXOS-1 from the nucleoli (S2I Fig) and found that four of them (except *snu-13* RNAi) improved stress resistance (Fig 2H), implying that the nucleolar RNA exosome may be a regulator of stress resistance. Further, using oxidative stress resistance as an example, we found that starved worms were extremely susceptible to stress (Figs 2I and S2J). And strikingly, RNA exosome inactivation restored stress resistance (Figs 2I and S2J) and also significantly increased the survival of fasted animals (Fig 2J). In addition, the TOR mutation and RNA exosome inactivation promoted stress resistance in a non-additive manner (S2K Fig), suggesting that they act in the same pathway to enhance cell maintenance. Altogether, we propose that organisms may sacrifice temporarily non-essential processes, such as the nucleolar RNA exosome, to facilitate cell maintenance and survival during starvation.

## Adaptive response to RNA exosome inactivation

Next, RNA-seq analysis was performed on *exos-8*(*RNAi*) worms to better characterize the cellular response to RNA exosome inactivation. WormCat, a tool designed specifically for *C. elegans*, was used to perform gene enrichment analysis, which provides enrichment data with both broad categories (Category One) and more specific categories (Categories Two and Three) [30]. Category One showed that *exos-8* RNAi-downregulated genes were involved in mRNA functions, transcription, and nucleic acid (Fig 3A), indicating a disruption in cellular RNA homeostasis. Intriguingly, RNA exosome inactivation also affected categories related to DNA and protein homeostasis (Fig 3A), suggesting that RNA perturbation may influence its upstream and downstream molecular processes. A more detailed examination of Categories Two and Three also revealed enrichment in groups related to DNA, RNA, and protein homeostasis, as well as an additional group of "Ribosome: EIF (Eukaryotic Initiation Factors)", which is indirectly connected with protein homeostasis (S3A and S3B Fig). These findings collectively support the notion that RNA exosome inactivation has extensive and negative effects on cellular nucleic acid and protein homeostasis.

Then, we analyzed genes induced by *exos-8* RNAi with the hypothesis that certain upregulated genes may represent the cellular adaptation to RNA exosome inactivation. Category One analysis indeed revealed a group of "stress response" genes (Fig 3B), confirming our earlier conclusion that cells adapt to RNA exosome inactivation by improving cellular maintenance. Categories Two and Three analysis showed that "stress response" genes were mainly involved in detoxification and pathogen stress response (S3C and S3D Fig). *exos-8* RNAi also induced gene groups associated with multiple essential cellular processes, including transmembrane transport, neuronal function, and extracellular material (Fig 3B), which may warrant further investigation. Transcriptomic data collectively revealed a gene map of the systemic stress response to RNA exosome inactivation, which can facilitate cellular maintenance during RNA stress.

## The RNA exosome does not regulate stress response genes directly or via nucleolus size

Next, we explored how RNA exosome inactivation promotes stress resistance. Generally, the RNA exosome associates with RNA molecules and regulates gene expression post-transcriptionally through direct degradation of mRNA [31]. However, our experimental evidence did not support this model, as RNAi of the exosome genes can induce the expression of multiple stress response reporters, including *gst-4p*::GFP, *hsp-4p*::GFP, *hsp-6p*::GFP, and *hsp-16.2p*::GFP (Figs 1A and S1C–S1E), which are transcriptional reporters driven by promoters only. To further corroborate this, we constructed several reporter strains for the oxidative stress gene *mtl-1*, including a transcriptional reporter (*mtl-1p*::GFP), a translational reporter (*mtl-1p*::MTL-1::GFP), and an ectopic expression reporter (intestinal *ges-1p*::MTL-1::GFP). *exos-8* RNAi increased the expression of both transcriptional and translational reporters (S3E and S3F Fig), but not the ectopic reporter (S3G Fig). Additionally, the induction of the *mtl-1* translational reporter was dependent on its transcription factor

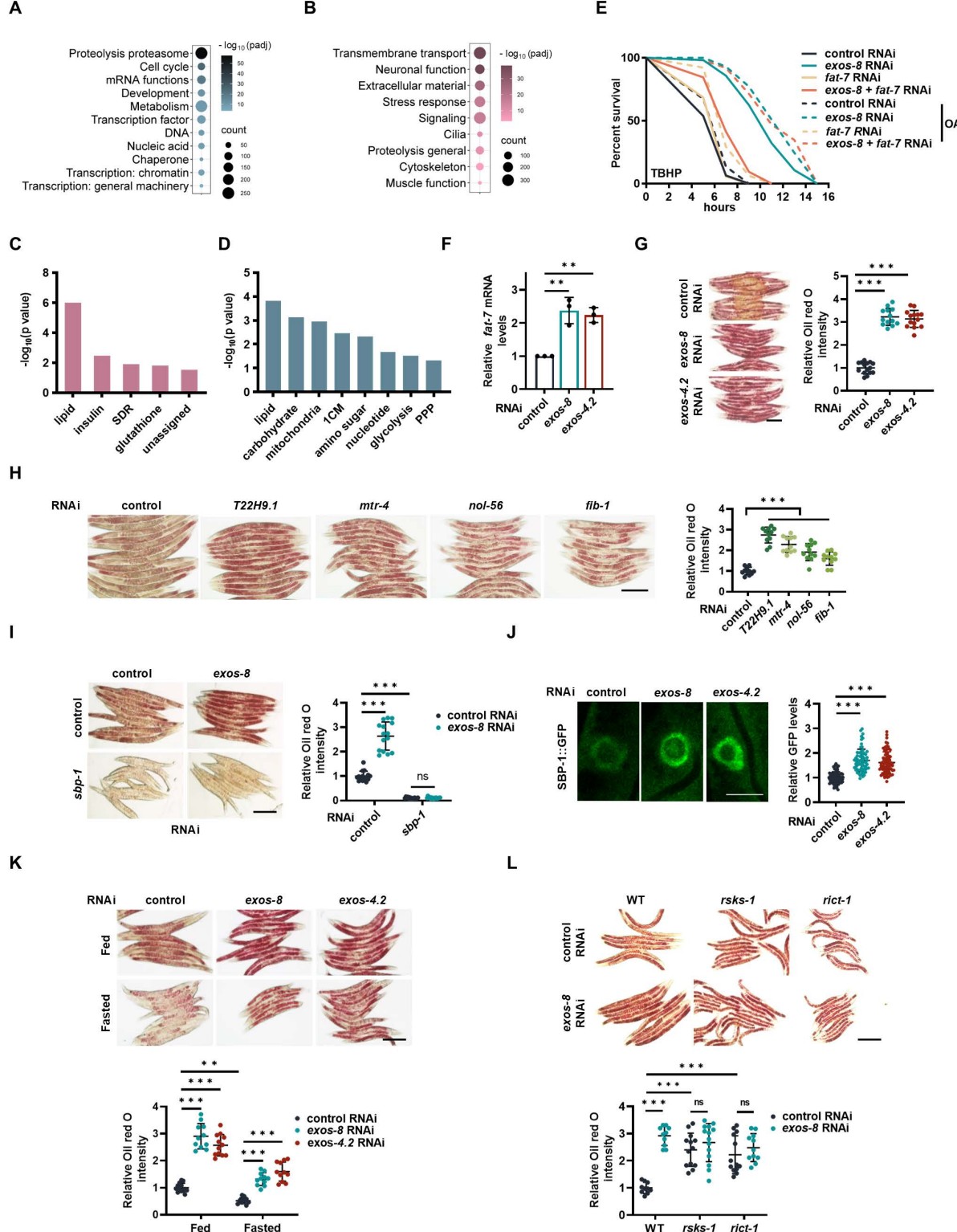

**Fig 3. RNA exosome inactivation promotes lipid biosynthesis. (A, B)** Functional classification of genes downregulated (A) and upregulated (B) by *exos-8* RNAi using broad functional categories (Category One classification in Wormcat). **(C, D)** Functional classification of metabolic genes upregulated (C) and downregulated (D) by *exos-8* RNAi, analyzed through specific functional categories (Category Two classification in Wormcat). SDR: short

PLOS Biology | https://doi.org/10.1371/journal.pbio.3003190  May 21, 2025
7 / 23

chain dehydrogenase; PPP: pentose phosphate pathway. **(E)** *fat-7* RNAi abolishes *exos-8* RNAi-induced oxidative stress resistance that is rescued by OA supplementation. **(F)** Effects of *exos-8* and *exos-4.2* RNAi on the mRNA expression of *fat-7*. One-way ANOVA with Dunnett's multiple comparisons test ($p = 0.0012$ for *exos-8* RNAi and $p = 0.0022$ for *exos-4.2* RNAi) ($n = 3$ experiments). **(G)** Left: Effects of *exos-8* and *exos-4.2* RNAi on fat accumulation measured by Oil red O staining. Scale bar = 200 μm. Right: Relative Oil red O intensity. One-way ANOVA with Dunnett's multiple comparisons test ($n = 14–15$ worms). **(H)** Left: Fat accumulation in animals treated with RNAi targeting nucleolar genes that affect the nucleolar occupancy of the RNA exosome. Scale bar = 200 μm; Right: Relative Oil red O intensity. One-way ANOVA with Dunnett's multiple comparisons test ($n = 11$ worms). **(I)** Left: Effects of *sbp-1* RNAi on *exos-8* RNAi-induced lipid accumulation, Scale bar = 200 μm; Right: Relative Oil red O intensity. Two-way ANOVA with Turkey's multiple comparisons test ($n = 14–15$ worms). **(J)** Effects of *exos-8* and *exos-4.2* RNAi on the nuclear occupancy of SBP-1::GFP in intestinal cells. Scale bar = 10 μm. Right: Relative GFP intensity in the nucleus. One-way ANOVA with Dunnett's multiple comparisons test ($n = 70-83$ cells). **(K)** Effects of *exos-8* and *exos-4.2* RNAi on lipid accumulation in well-fed and starved animals. Scale bar = 200 μm. Right: Relative Oil red O intensity. Two-way ANOVA with Turkey's multiple comparisons test (**$p = 0.0035$) ($n = 11–12$ worms). **(L)** Effects of the *rsks-1* and *rict-1* mutations on *exos-8* RNAi-induced lipid accumulation. Scale bar = 200 μm. Right: Relative Oil red O intensity. Two-way ANOVA with Turkey's multiple comparisons test ($n = 10–13$ worms). Data are presented as mean ± SD. **$p < 0.01$, ***$p < 0.001$. S1 Table provides all repeats and statistical analyses of the survival experiments, where Repeat 1 of each experiment was used for generating the graphs. The numerical data presented in this figure can be found in S1 Data.

DAF-16 (S3F Fig), suggesting that RNA exosome inactivation promotes *mtl-1* expression via a transcriptional mechanism. Although we did not test all stress genes, it is probable that the RNA exosome does not regulate them directly.

Recently, nucleolus size was found to be a regulator of stress responses and aging [32,33]. Given that the RNA exosome is primarily located in the nucleolus, it might influence stress responses through nucleolus size. To test this, we used the nucleolar marker FIB-1::mCHERRY[27] and found that exosome genes RNAi did not affect nucleolus size (S3H Fig). Additionally, since a smaller nucleolus is typically associated with enhanced stress resistance, we examined *ncl-1* mutants, which have enlarged nucleoli and suppress longevity in several long-lived models [33]. Notably, even in *ncl-1* mutants, RNA exosome suppression still resulted in increased stress resistance (S3I Fig). These findings suggest that RNA exosome inactivation does not regulate stress response via changes in nucleolus size.

### Lipid metabolism promotes cellular maintenance in response to RNA exosome inactivation

As RNA exosome inactivation promotes multiple stress responses, we postulated that a shared mechanism is responsible for this phenotype. A large number of metabolic genes were either upregulated or downregulated by *exos-8* RNAi (S5 Table), indicating extensive metabolic reprogramming in response to RNA exosome inactivation. We speculated that these metabolic changes might act downstream of the RNA exosome for two key reasons: (1) numerous metabolic pathways or metabolites have been implicated in the regulation of stress responses in *C. elegans*; and (2) the RNA exosome responds to nutrient stress to modulate stress resistance (Fig 2), thus metabolic changes might function to coordinate metabolic adaptation and stress resistance in response to RNA exosome inactivation.

It is currently unclear whether and how RNA homeostasis may impact nutrient metabolism. To shed light on this, we used WormCat to analyze metabolic genes in greater depth and discovered that the *exos-8* RNAi-upregulated metabolic genes were primarily associated with lipid and insulin metabolism (Fig 3C), and the downregulated genes were involved in the metabolism of lipid, carbohydrate, mitochondria, 1CM, and amino sugar (Fig 3D). To identify the precise metabolic pathway responsible for the RNA exosome response, we utilized RNAi or mutants to inactivate key genes of the aforementioned pathways and examined oxidative stress resistance. *fat-7* was thus identified, which encodes the enzyme responsible for generating oleic acid (OA), an unsaturated FA essential for various stress resistance in *C. elegans* [34–38]. Likewise, *fat-7* RNAi impaired oxidative stress resistance in *exos-8(RNAi)* worms (Fig 3E), which can be completely rescued by OA supplementation (Fig 3E), indicating that OA is also critical in triggering stress resistance when the RNA exosome is inactivated.

The enrichment of the "lipid" category and the identification of FAT-7 suggest that lipid metabolism may be regulated by the RNA exosome. The qPCR results confirmed the elevated expression of *fat-7* (Fig 3F). The RNA-seq data showed that the majority of FA metabolic genes, including both FA synthetic and catabolic genes, were also upregulated by *exos-8* RNAi (S4A Fig), indicating an increase in lipid turnover. Then, using Oil red O and Nile red, two independent staining

methods for cellular neutral lipids, we observed a significant increase in fat accumulation following the inactivation of the RNA exosome (Figs 3G, S4B, and S4C). Similar fat accumulation phenotypes were also observed after RNAi of nucleolar genes that depleted nucleolar GFP::EXOS-1 (Fig 3H). Next, we tested the upstream transcription factors involved in FA metabolism. The mutation or RNAi knockdown of MXL-3 [39], NHR-49 [5,6,40], and HLH-11 [7], transcription factors for FA oxidation, had no effects on lipid accumulation in *exos-8(RNAi)* worms (S4D–S4F Fig), whereas RNAi knockdown of SBP-1/SREBP1 [41], which regulates FA biosynthesis, abolished fat accumulation (Fig 3I). RNA exosome inactivation also increased the nuclear occupancy and protein levels of SBP-1 (Figs 3J and S4G). These results collectively indicate that RNA exosome inactivation promotes lipid accumulation, with the FA OA serving as a key mediator of enhanced stress resistance.

Having established a critical role for the RNA exosome in lipid accumulation, we wondered whether this regulation occurs during nutrient deprivation. Indeed, starvation caused a dramatic decrease in fat accumulation, but RNA exosome inactivation largely counteracted this effect and preserved more lipids (Figs 3K and S4H). Additionally, TOR inhibition and RNA exosome inactivation had no additive effects on lipid accumulation (Fig 3L), consistent with our finding that TOR is a regulator of the RNA exosome. Together, we propose that nucleolar RNA surveillance may be considered a temporarily non-essential program that can be disabled. Its inactivation may further serve as a key signal to generate more lipids, which can be mobilized for stress resistance during starvation.

## 1CM mediates the RNA exosome response

The RNA exosome does not appear to be directly related to fat metabolism, implying the existence of an intermediary mechanism connecting the two processes. As such, we examined cellular processes related to the function of the RNA exosome. The expression of *atic-1* and *F38B6.4*, two major enzymes catalyzing purine production, were upregulated in response to *exos-8* RNAi (Figs 4A and S5A), while *tyms-1*, which converts dUMP to dTMP for DNA synthesis, was downregulated (Figs 4A and S5A). This suggests that nucleotide metabolic enzymes may be adaptively regulated for RNA biosynthesis. Intriguingly, these enzymes all utilize one-carbon units from 1CM to produce nucleotides. For example, purine biosynthetic enzymes employ 1CM intermediate 10-formyl-THF (10-fTHF) for nucleotide synthesis, which connects 1CM to nucleotide synthesis. 1CM is at the crossroads of several metabolic pathways, the center of which is the folate cycle and the methionine cycle [42] (Fig 4A). The 1CM genes were enriched in the metabolic pathways that were suppressed by *exos-8* RNAi (Fig 3D), and their downregulation was confirmed by qPCR (Fig 4B). The suppression of 1CM genes and concomitant induction of purine biosynthetic genes may indicate a metabolic shift from 1CM toward nucleotide biosynthesis, allowing the intermediate 10-fTHF to enter nucleotide biosynthesis. Supporting this notion, the expression of *dao-3*, which produces 10-fTHF from the 1CM metabolite 5, 10-methylene-THF (5, 10-meTHF), is upregulated by *exos-8* RNAi (Figs 4A and S5A). Altogether, we propose that in response to disrupted RNA surveillance, cells rewire 1CM and nucleotide metabolism to promote nucleotide biosynthesis, which may mitigate RNA stress.

Given the importance of 1CM in cellular metabolism, metabolic reprogramming in 1CM could have a great impact on cell function. Thus, we inactivated enzymes in related pathways and examined fat accumulation in response to *exos-8* RNAi. This approach identified the only candidate *sams-1*, encoding the key enzyme of 1CM that produces the major methyl donor S-adenosylmethionine (SAMe) in cells (Fig 4A). The *sams-1* mutation increased fat accumulation, and its effect was not additive to RNA exosome inactivation (Figs 4C and S5B), suggesting that the downregulation of *sams-1* mediates fat buildup in response to RNA exosome inactivation. Unexpectedly, the *sams-1* mutation did not improve but abolished oxidative stress resistance in *exos-8(RNAi)* animals (Fig 4D). We speculated that *sams-1* mutants may be susceptible to oxidative stress due to inherent defects. Such a perplexing finding encouraged us to explore another stress resistance assay (mitochondrial stress), and we observed that the *sams-1* mutation enhanced mitochondrial stress resistance and exhibited no additive effects with RNA exosome inactivation (Figs 4E and S5C), verifying that *sams-1* inhibition mediates the cellular response to RNA exosome disturbance.

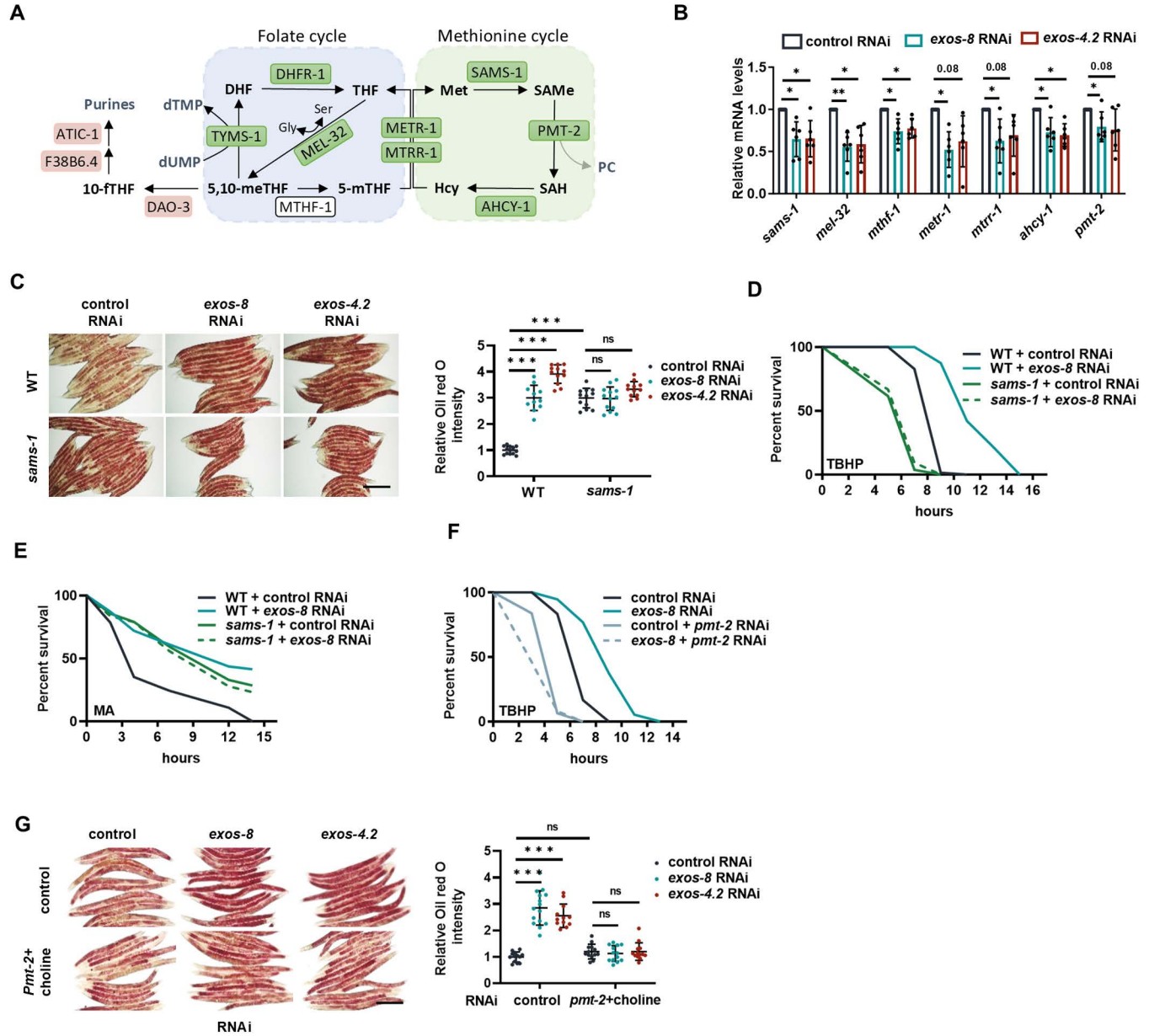

**Fig 4. The 1CM metabolite SAMe mediates the RNA exosome response. (A)** Schematics of 1CM and the expression of related enzymes in *exos-8(RNAi)* worms. Enzymes upregulated by *exos-8* RNAi in RNA-seq results are shown in red boxes, while downregulated enzymes are shown in green boxes. **(B)** Effects of *exos-8* and *exos-4.2* RNAi on 1CM gene expression. Multiple *t* test with correction for multiple comparisons using the Holm–Sidak method (**$p = 0.0099$ for *mel-32*, *$p = 0.0376/0.0421$ for *sams-1*, 0.03 for *mel-32*, 0.0376/0.0281 for *mthf-1*, 0.0154 for *metr-1*, 0.0376 for *mtrr-1*, 0.0376/0.0174 for *ahcy-1*, 0.0376 for *pmt-2*) ($n = 6$ experiments). **(C)** Effects of the *sams-1* mutation on *exos-8* and *exos-4.2* RNAi-induced lipid accumulation. Scale bar = 200 μm. Right: Relative Oil red O intensity. Two-way ANOVA with Turkey's multiple comparisons test ($n = 11–12$ worms). **(D, E)** Effects of the *sams-1* mutation on *exos-8* RNAi-induced oxidative stress resistance (D) and mitochondrial stress resistance (E). **(F)** Knockdown of *pmt-2* abolishes *exos-8* RNAi-induced oxidative stress resistance. **(G)** Left: Effects of *pmt-2* RNAi with choline supplementation on RNA exosome inactivation-induced lipid accumulation. Scale bar = 200 μm. Right: Relative Oil red O intensity. Two-way ANOVA with Turkey's multiple comparisons test ($n = 12–13$ worms). Data are presented as mean ± SD. *$p < 0.05$, **$p < 0.01$, ***$p < 0.001$. S1 Table provides all repeats and statistical analyses of the survival experiments, where Repeat 1 of each experiment was used for generating the graphs. The numerical data presented in this figure can be found in S1 Data.

Downregulation of *sams-1* may affect its product SAMe and upstream metabolites (Fig 4A). We therefore used Oil red O staining as a simple readout to examine the involvement of SAMS-1-related 1CM processes in fat accumulation. The metabolic pathways of serine, glycine, and methionine are upstream of SAMS-1 (Fig 4A); however, the addition of these amino acids had little or no effects on the lipid buildup in either WT or RNA exosome-inactivated worms (S5D–S5F Fig), indicating that amino acids may not account for the RNA exosome effects. RNAi inactivation of upstream 1CM genes had little or no effects on lipid accumulation either (S5G Fig). Thus, 1CM pathways upstream of SAMS-1 may not be involved in the cellular responses to RNA exosome inactivation.

Next, we asked whether the methyl donor SAMe mediates the effects of RNA exosome inactivation. The enzyme PMT-2 catalyzes the conversion of SAMe into the membrane lipid phosphatidylcholine (PC). SAMe levels are thus balanced by the activity of SAMS-1 and PMT-2 (Fig 4A). We reasoned that if SAMe reduction accounts for lipid buildup, then PMT inhibition, which increases SAMe levels, would abolish the RNA exosome effects. As expected, *pmt-2* RNAi indeed inhibited lipid accumulation and stress resistance in RNA exosome-inactive animals (Figs 4F and S5H). The *pmt-2(RNAi)* worms appear to be small and thin, likely due to drastically reduced PC levels. To avoid confounded results, we further combined *pmt-2* RNAi with choline supplementation, the precursor of PC synthesis, to alter SAMe levels without reducing PC levels. This treatment restored the morphology of *pmt-2(RNAi)* worms and, at the same time, inhibited lipid accumulation in animals with inactivated RNA exosome (Fig 4G). These results suggest that SAMe reduction mediates the cellular response to RNA exosome inactivation. Together, we propose that in response to RNA exosome inactivation, a shift from 1CM toward purine metabolism coordinately controls fat metabolism and nucleotide metabolism.

### Translation inhibition preserves energy in response to RNA exosome inactivation

How does SAMe mediate the RNA exosome effects? Previous research has shown that *sams-1* inactivation promotes fat synthesis by decreasing PC levels [43]. The addition of PC indeed abolished fat accumulation in *sams-1* mutants (S6A Fig), but it did not affect fat buildup in *exos-8(RNAi)* worms (S6B Fig), suggesting that *sams-1* may regulate fat metabolism independent of PC when the RNA exosome is inhibited.

We next explored the PC-independent mechanism of fat buildup. Since RNA exosome inactivation enhances fat accumulation in response to nutrient deprivation, we speculated that it might inhibit certain non-essential energy-consuming processes for energy preservation. This idea encouraged us to consider the translation process, one of the most energy-intensive cellular processes [44]. Moreover, when the nucleolar RNA exosome is perturbed, aberrant rRNA accumulates and may act as a "no more translation" signal. Supporting this idea, *exos-8* RNAi downregulated the category "ribosome: EIF" (S3B Fig), which encodes eukaryotic translation initiation factors (S6C Fig). As such, we measured cellular translation and discovered that RNA exosome inactivation significantly inhibited translation (Figs 5A and S6D). This suppressive effect was not additive to TOR inhibition (Fig 5B), suggesting that these molecules regulate translation in a linear way. SAMS-1 has been reported to regulate global translation in *C. elegans* and mammals [45,46]. We found that the *sams-1* RNAi indeed suppressed translation and that this effect was not additive to RNA exosome inactivation (Fig 5C). More importantly, SAMe elevation treatment (*pmt-2* RNAi with choline addition) reversed translation in *exos-8(RNAi)* worms (Fig 5D). These findings collectively suggest that in response to RNA exosome inactivation, cells inhibit 1CM to reduce SAMe, which then inhibits translation.

Further, we dissect the role of translation inhibition in lipid metabolism and stress resistance. EIF-2Bβ and EIF-2β are components of the translation-initiation complex whose expression was inhibited by *exos-8* RNAi (S6C Fig). RNAi knockdown of *eif-2Bβ* and *eif-2β* increased lipid accumulation (Figs 5E and S6E) and stress resistance (Figs 5F and S6F) in a manner that was not additive to RNA exosome perturbation. The non-additive effects between the RNA exosome and translation inhibition on enhanced stress resistance were confirmed by *exos-10* mutants and a translation inhibitor cycloheximide (S6G Fig). We also showed that starvation profoundly suppressed cellular translation (S6H Fig), which benefits lipid preservation (Fig 5G) and stress resistance (Fig 5H). Altogether, these findings suggest a model in which

PLOS Biology

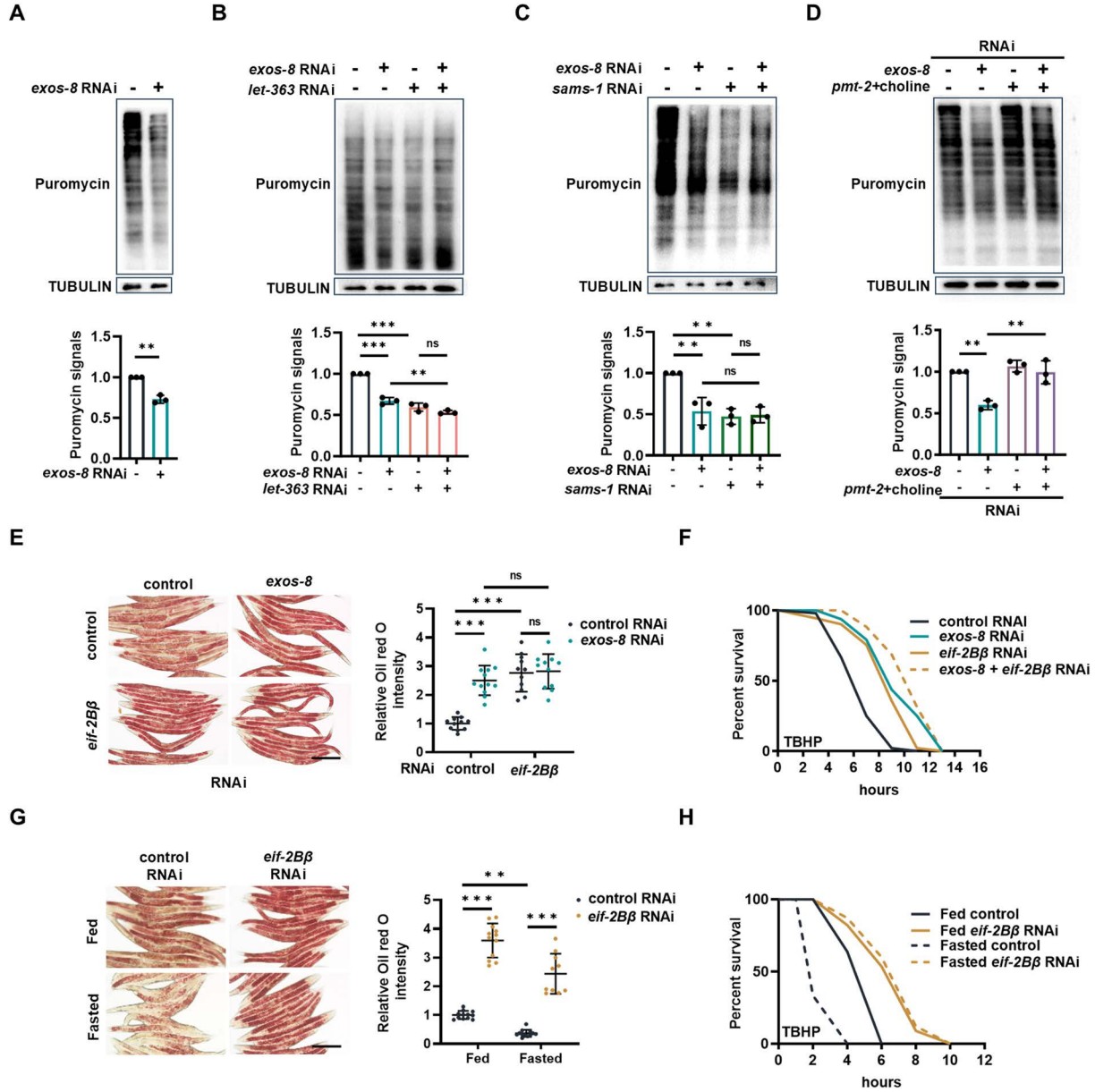

**Fig 5. Translation inhibition preserves energy in response to RNA exosome inactivation. (A)** Upper: Effects of *exos-8* RNAi on translation. Lower: Quantification of translation. Paired two-tailed *t* test (**\*\*p* = 0.0095) (*n* = 3 experiments). **(B)** Upper: *exos-8* RNAi and *let-363/Tor* RNAi have no additive effects on translation inhibition. Lower: Quantification of translation. Two-way ANOVA with Turkey's multiple comparisons test (**\*\*p* = 0.0049) (*n* = 3 experiments). **(C)** Upper: *exos-8* RNAi and *sams-1* RNAi have no additive effects on translation inhibition. Lower: Quantification of translation. Two-way ANOVA with Turkey's multiple comparisons test (**\*\*p* = 0.0033/0.0014) (*n* = 3 experiments). **(D)** Upper: SAMe elevation treatment (*pmt-2* RNAi + choline) reverses *exos-8* RNAi-induced translation inhibition. Lower: Quantification of translation. Two-way ANOVA with Turkey's multiple comparisons test (**\*\*p* = 0.0015/0.0017) (*n* = 3 experiments). **(E)** Left: Effects of *eif-2Bβ* RNAi on *exos-8* RNAi-induced lipid accumulation. Scale bar = 200 μm. Right: Relative Oil red O intensity. Two-way ANOVA with Turkey's multiple comparisons test (*n* = 11−12 worms). **(F)** Effects of *eif-2Bβ* RNAi on *exos-8* RNAi-induced oxidative stress resistance. **(G)** Left: Effects of *eif-2Bβ* RNAi on lipid accumulation in well-fed and starved animals. Scale bar = 200 μm. Right: Relative Oil red O intensity. Two-way ANOVA with Turkey's multiple comparisons test (**\*\*p* = 0.0071) (*n* = 10−12 worms). **(H)** Effects of *eif-2Bβ* RNAi on oxidative stress resistance in well-fed and starved animals. Data are presented as mean ± SD. *\*p* < 0.05, **\*\*p* < 0.01, ***\*\*\*p* < 0.001. S1 Table provides all repeats and statistical analyses of the survival experiments, where Repeat 1 of each experiment was used for generating the graphs. The numerical data presented in this figure can be found in S1 Data.

RNA exosome inactivation adaptively inhibits translation via 1CM, thereby conserving energy for lipid buildup and cellular maintenance.

### RNA exosome inactivation promotes longevity in dietary restriction (DR) animals

Enhanced stress resistance is typically associated with longevity, but RNA surveillance should be crucial for long-term survival. How RNA exosome inactivation may affect life span is a very intriguing question. We found that *exos-8* RNAi shortened the life span of WT animals (Fig 6A), indicating that RNA surveillance by the RNA exosome is essential for normal life span and that enhanced stress resistance is insufficient to promote longevity. Since the developmental stage necessitates active and rigorous transcription and translation that may require RNA surveillance in particular, we asked whether adult-initiated exosome inactivation could mitigate its deleterious effects, thereby making it more advantageous

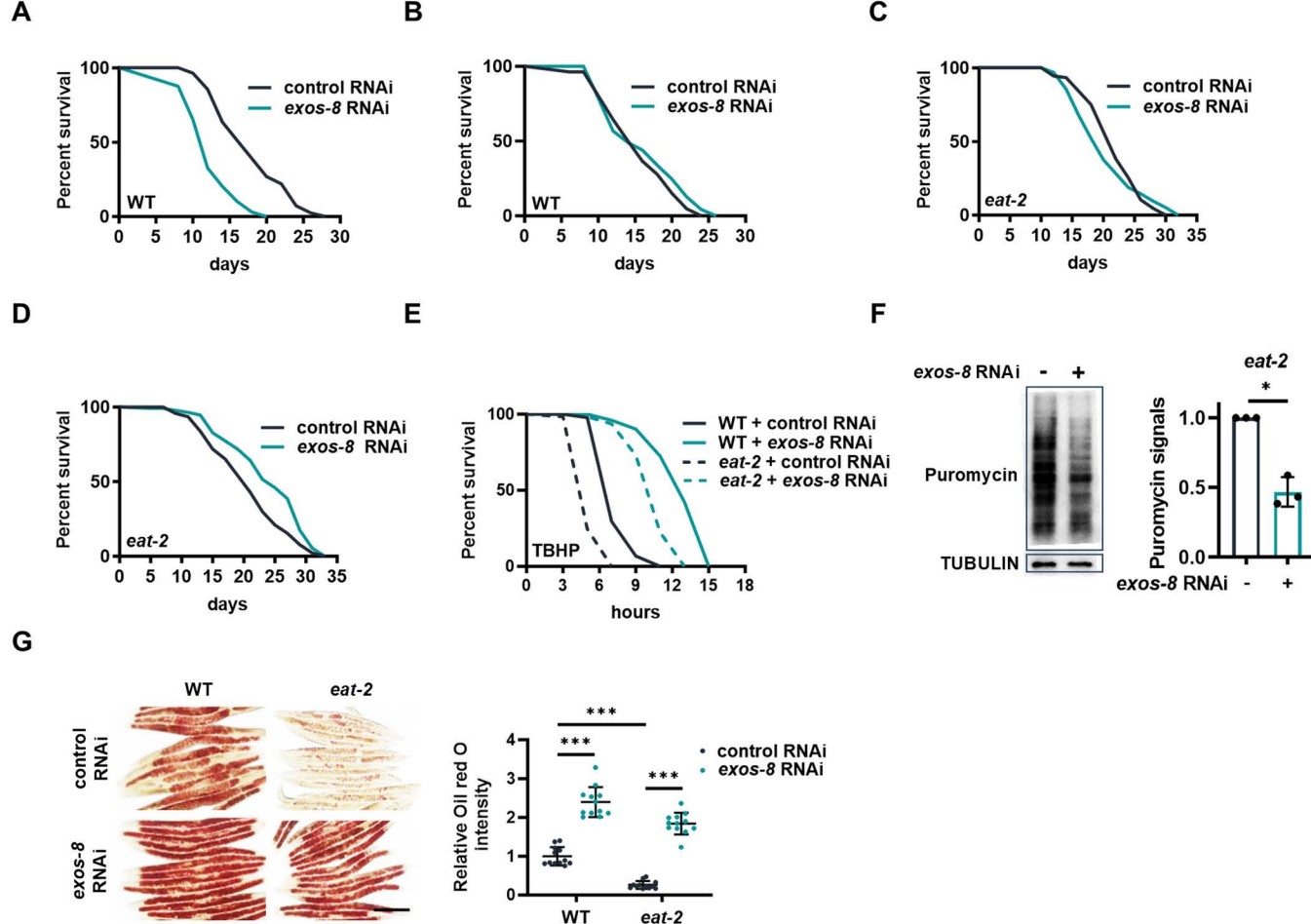

**Fig 6. Regulation of life span by the RNA exosome. (A, B)** Effects of *exos-8* RNAi from birth (A) or after development (B) on the life span of WT animals. **(C, D)** Effects of *exos-8* RNAi from birth (C) or after development (D) on the life span of *eat-2* mutants. **(E)** Effects of *exos-8* RNAi on the oxidative stress resistance of *eat-2* mutants. **(F)** Left: *exos-8* RNAi reduced the translation rate in *eat-2* mutants. Right: Quantification of translation. Paired two-tailed *t* test (*$p = 0.013$) ($n = 3$ experiments). **(G)** Left: Effects of *exos-8* RNAi on lipid accumulation in *eat-2* mutants. Scale bar = 200 μm. Right: Relative Oil red O intensity. Two-way ANOVA with Turkey's multiple comparisons test ($n = 12$ worms). Data are presented as mean ± SD. *$p < 0.05$, ***$p < 0.001$. S1 and S2 Tables provide all repeats and statistical analyses of the survival experiments, where Repeat 1 of each experiment was used for generating the graphs. The numerical data presented in this figure can be found in S1 Data.

for longevity. Post-developmental *exos-8* RNAi indeed enhanced stress resistance (S7A and S7B Fig) without reducing normal life span (Fig 6B), which appears to be superior to RNAi initiated at birth. We propose that the negative and positive effects may counteract each other when the RNA exosome is inactivated post-developmentally.

Next, we wondered how RNA exosome inactivation may affect the life span of a long-lived dietary restriction (DR) model with the *eat-2* mutation [47], as inhibiting RNA exosome benefits starvation survival. *exos-8* RNAi from birth did not reduce the life span of *eat-2* mutants (Fig 6C) and even modestly extended life span when treated post-developmentally (Fig 6D), indicating that RNA exosome inactivation is particularly advantageous for longevity in the context of DR.

Why does this occur? Like starved animals, DR animals also suffer from nutritional deficiency, which may be detrimental to cell maintenance. Consistent with this notion, *eat-2* mutants indeed exhibited impaired stress resistance, which was reversed by *exos-8* RNAi (Fig 6E). Moreover, the restoration of stress resistance was associated with a reduction in translation (Fig 6F) and the preservation of fat in DR animals (Fig 6G). These findings further suggest that RNA exosome inactivation is particularly advantageous for organism fitness during DR.

We also tested another longevity model with the insulin/IGF-1 receptor *daf-2* mutation. The results of *daf-2* mutants were comparable to those of WT animals. *exos-8* RNAi still improved the oxidative stress resistance of *daf-2* mutants (S7C Fig), but it significantly or moderately reduced life span when administered at birth or after development (S7D and S7E Fig). These findings suggest that the life span-extending effect of RNA exosome inactivation may be specific to nutrient deprivation.

## Discussion

In the current study, we find that the nucleolar RNA exosome is depleted by nutrient scarcity. This may act as a signal to rewire 1CM, resulting in translation inhibition to preserve fat for cellular maintenance (Fig 7). It is generally accepted that the RNA exosome is vital for proper cellular function. Mutations or defects in RNA exosome components have been associated with a variety of human disorders [21,22]. However, we find that RNA exosome inactivation has a positive effect. We propose that cells have evolved a surveillance or adaptive response to cope with RNA exosome defects. The enhancement of stress resistance is such an adaptation to endure RNA disorders, which allows the maintenance of cellular functions and the restoration of RNA homeostasis. Notably, previous research has documented that DNA damage

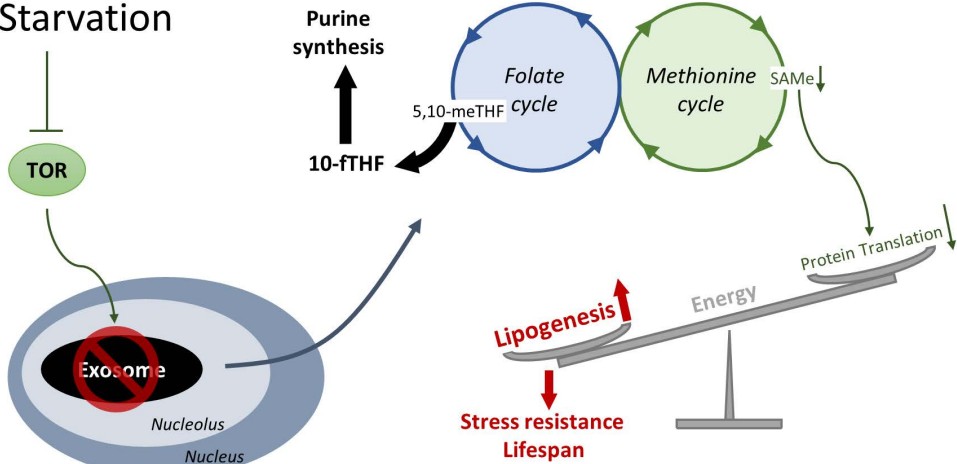

**Fig 7. Model of starvation adaptation controlled by the RNA exosome.** In response to starvation, the nucleolar RNA exosome is depleted in a way that may depend on TOR. Cells respond by rewiring 1CM to coordinately increase nucleotide metabolism and decrease the metabolite SAMe, which inhibits global translation. Thus, energy is conserved for fat synthesis, which improves cellular maintenance and survival in the face of starvation.

in *C. elegans* can evoke a similar stress response to facilitate cellular endurance [48,49], suggesting that enhancement of cellular preservation could serve as a general mechanism for coping with defects in nucleotide homeostasis.

In *C. elegans*, the intestine functions as the primary effector tissue for metabolic and stress responses, where stress-response genes exhibit predominant expression, and lipid accumulation predominantly occurs. We therefore propose that the observed intestinal alterations in stress-response gene expression and lipid staining patterns may underlie the stress-resistant phenotype of RNA exosome-deficient animals. However, it should be noted that while we observed nucleolar expression and regulation of the RNA exosome in intestinal cells, this does not preclude potential involvement of RNA exosome activity in other tissues for systemic metabolic and stress regulation. The RNA exosome exhibits broad tissue distribution, with documented expression in epidermal cells, for instance [25,26]. Consequently, key unresolved questions persist regarding whether tissue-specific RNA exosome activity coordinates organism-wide metabolic/stress responses and whether cell non-autonomous regulatory mechanisms exist—both requiring further investigation.

The RNA exosome is a complex with exoribonuclease activity, but it also exhibits endoribonuclease activity. In other organisms, such as yeast, the DIS-3 catalytic subunit demonstrates both endoribonuclease and exoribonuclease activities, while the EXOS-10 subunit is restricted to exoribonuclease function [50–52]. The *C. elegans* EXOS-10 sub-unit shares conserved functional domains with its orthologs, suggesting a conserved exoribonuclease function [25]. We confirmed the exoribonuclease activity of EXOS-10 in the nucleolus, where it participates in rRNA processing. In *exos-10* mutants, 18S pre-rRNA with ITS1 regions accumulates, indicating impaired processing. Similar to RNAi knockdown of core exosome subunits, *exos-10* mutants displayed lipid accumulation and enhanced stress resistance. These findings suggest that loss of exoribonuclease activity in the RNA exosome is sufficient to trigger these pheno-types. Currently, it remains unclear whether *C. elegans* DIS-3 possesses both endoribonuclease and exoribonuclease activities. A comprehensive analysis of the DIS-3 subunit is necessary to address this question, which could provide critical insights into whether endoribonuclease activity contributes to the RNA exosome's role in regulating stress resistance.

Notably, rewired 1CM coordinates multiple adaptations to RNA exosome perturbation, including nucleotide metabolism, translation, and fat metabolism (Fig 7), suggesting a central role of 1CM in response to RNA exosome disturbance. Meta-bolic shift from 1CM to nucleotide biosynthesis is important for rRNA production [53], which is consistent with a crucial role for the nucleolar RNA exosome in rRNA quality control and may facilitate adaptation to aberrant rRNA buildup. Meanwhile, translation inhibition can mitigate ribosome stress caused by rRNA surveillance deficiency that results from RNA exosome perturbation. In addition, energy is preserved for stress resistance, which, as previously indicated, may facilitate cellular endurance to allow the repair and recovery of RNA surveillance machinery. Thus, multiple adaptive responses are effi-ciently coordinated by 1CM. How 1CM is regulated by the RNA exosome warrants further investigation.

Cells and organisms are continuously confronted with various environmental and cellular stresses. Thus, tradeoff mechanisms are critical for balancing energy demands and reallocating energy to essential processes [54]. This is espe-cially critical during starvation when the external energy supply is limited. We find that the nucleolar RNA exosome is inac-tivated by nutrient deprivation, which suggests a tradeoff between RNA surveillance and nutrient scarcity adaptation. Why can cells endure RNA exosome inactivation? The function of the RNA exosome is crucial for long-term fitness, including life span. On the contrary, the temporal inhibition of the RNA surveillance mechanism is tolerable to cells because RNA molecules undergo a rapid turnover. In addition, starvation is associated with the suppression of rRNA biogenesis [55], which is advantageous for the tolerance of RNA exosome inactivation. Consequently, the nucleolar RNA exosome can be temporarily sacrificed, which enables the tradeoff between rRNA surveillance and starvation adaptation.

In recent decades, studies have focused on the direct regulation of fat metabolic enzymes. Numerous elegant works illuminate how cells detect nutrient scarcity and regulate fat metabolism transcriptionally. Our work provides another regulatory layer for energy homeostasis during starvation that integrates additional cellular processes. However, it should

be noted that our research merely exemplifies the intricate interplay that exists between metabolism and other cellular processes. There may exist other unidentified non-essential processes that require cessation to enable starvation adaptation.

In summary, we describe a mechanism that enables energy reallocation during nutrient scarcity, which facilitates cellular maintenance and favors organismal survival. This response is critically governed by the activity of the RNA exosome, suggesting a new regulatory role for RNA surveillance machinery and a tradeoff between RNA homeostasis and nutrient adaptation. Based on the conserved function of the RNA exosome, we propose that this response could be an ancestral adaptation to nutrient scarcity that is shared by mammalian cells.

## Materials and Methods

### *C. elegans* strains and maintenance

*C. elegans* were cultured on standard nematode-growth medium (NGM) seeded with the indicated bacteria [56]. The following strains were provided by Caenorhabditis Genome Center: the WT N2 Bristol, CB1370[*daf-2(e1370)*], KQ1366[*rict-1(ft7)*], RB1206[*rsks-1(ok1255)*], VC2428[*sams-1(ok2946)*], TJ375[*hsp-16.2p::gfp*], STE68[*nhr-49(nr2041)*], AU78[*T24B8.5p::gfp*], CL2166[*gst-4p::gfp*], SJ4005[*hsp-4p::gfp*], SJ4100[*hsp-6p::gfp*], DA465[*eat-2(ad465)*], and RB1588[*mxl-3(ok1947)*]. KQ377[*sbp-1::gfp*] was provided by Professor Bin Liang. The strains containing the following reporters were constructed by our own lab: *mtl-1p::mtl-1::gfp* [57], *mtl-1p::gfp*, *ges-1p::mtl-1::gfp*, *exos-8::mcherry*.

### RNAi treatment

HT115 bacteria with specific dsRNA-expression plasmids [58] were grown overnight at 37 °C in LB containing 100 μg/ml carbenicillin and seeded onto NGM plates containing 5 mM IPTG. RNAi was induced at 25 °C for 24 h. L1 worms were introduced to the plates to knock down the indicated genes. The efficacy of RNAi knockdown was listed in S3 Table.

### Stress resistance assays

Stress resistance assays were performed as previously described [48] with some modifications. Stress resistance experiments were conducted using day 1 adults, except in cases involving *exos-10* mutants, where L4-stage worms were used. For TBHP resistance, worms were transferred to NGM plates supplemented with 12.6 mM TBHP for survival analysis. For MA resistance, worms were transferred to NGM plates supplemented with 50 mM MA for survival analysis. For DTT resistance, worms were transferred to NGM plates supplemented with 20 mM DTT for survival analysis. For the heat shock resistance assay, worms were cultured at 35°C for survival analysis.

### Lipid staining

Oil red O staining was performed as previously described [34,59]. Day 1 adult worms were collected and fixed in 60% isopropanol for 5 min. Worms were washed and collected for staining overnight with a freshly prepared Oil red O working solution. Following that, worms were washed twice with M9 buffer, mounted on slides, and imaged using bright-field illumination. For each experimental group, approximately 10–15 animals were randomly selected for microscopic imaging and quantitative analysis, which was performed using ImageJ 1.53e software (NIH). Each experiment was repeated at least three times, and the quantitative results were derived from the statistical analysis of 1–2 replicates.

### Fluorescent microscopy

Fluorescent analysis was performed as previously described [60]. To assess fluorescence, day 1 adult worms were immobilized with 1 mM levamisole and mounted on slides. The *hsp-6p::gfp* reporter was activated by exposure to 2

µM antimycin, while the *hsp-16.2p::gfp* reporter was induced by heat stress at 35°C. Fluorescent microscopic images were captured using Nikon NIS-Elements software or Leica LAS X software. The nucleolar localization patterns of GFP::EXOS-1, DIS-3::mCHERRY, and EXOS-8::mCHERRY were semi-quantified in the intestinal cells. The fluorescence intensity was measured in whole animals using ImageJ 1.53e software (NIH). For each experimental group, approximately 10–20 animals were randomly selected for microscopic imaging. The quantitative results were derived from statistical analysis of 1–2 replicates (for fluorescence intensity experiments) or pooled data from all replicates (for protein nucleolar localization and nucleolar size experiments).

## Western blot

Day 1 adult worms were collected and sonicated in RIPA buffer containing protease inhibitors (Sigma) before boiling and loading. Antibodies against GFP (Santa Cruz, SC-9996), puromycin (Sigma, MABE343), and TUBULIN (Sigma, T9026) were used. Images were quantified using ImageJ 1.53e software. Each experiment was repeated 3–4 times, and data from all replicates were pooled for statistical analysis.

## Detection of translation levels

Puromycin incorporation was used to monitor translation levels in a nonradioactive manner as previously described [61]. Worms were collected and washed twice with S buffer. Then, worms were incubated in S buffer containing 0.5 mg/ml puromycin and bacteria for 4 h at 200 rpm. After that, worms were harvested for western blot analysis using a monoclonal anti-puromycin antibody.

## cRT-PCR

Circular reverse transcription PCR (cRT-PCR) was performed as previously described [27]. Total RNA was isolated from L4 stage worms using TRIzol solution. Two micrograms of total RNA was circularized by T4 RNA Ligase 1 (NEB) and then purified by TRIzol reagent followed by isopropanol precipitation. The circularized RNA was reverse transcribed via the GoScript Reverse Transcription System (Promega). PCR was performed using a Rapid Taq Master Mix (Vazyme). The primers for cRT-PCR were listed in S4 Table.

## Quantitative RT-PCR

The quantitative real-time PCR (RT-PCR) was conducted as previously described [62]. Day 2 adult worms were collected, washed in M9 buffer, and then homogenized using TRIzol reagent. RNA extraction was performed according to the manufacturer's instructions. After DNA contamination was digested with DNase I, RNA was reverse-transcribed to complementary DNA using the RevertAid First Strand Complementary DNA Synthesis Kit (Thermo Fisher Scientific). SYBR Green was used for quantitative PCR, and data were obtained using Bio-Rad's CFX Maestro Software. The expression of *snb-1* was utilized to normalize samples. Each experiment was repeated at least three times, and data from all replicates were pooled for statistical analysis. qPCR primers were listed in S4 Table.

## Compound supplementation

OA was dissolved in dimethyl sulfoxide (DMSO) to generate a 300 mM stock solution, which was added onto the surface of NGM plates to create a working concentration of 300 µM. PC was dissolved in DMSO to generate a 17 mM stock solution, which was added onto the surface of NGM plates to create a working concentration of 50 nM [63]. Choline was dissolved in ddH$_2$O to generate a 1M stock solution, which was added onto the surface of NGM plates to create a working concentration of 7.5 mM. Amino acids were dissolved in ddH$_2$O to generate 1 M stock solutions, which were added onto the surface of NGM plates to create a working concentration of 500 µM.

## Lifespan analysis

Synchronized L1 worms were placed onto NGM plates with the indicated *Escherichia coli* strains. Worms were transferred every day during the reproductive period. Worms that died of vulva bursts, bagging, or crawling off the plates were censored.

## RNA sequencing analysis

Total RNA from day 2 stage worms was extracted using TRIzol reagent. RNA integrity was assessed using the RNA Nano 6000 Assay Kit of the Bioanalyzer 2100 system (Agilent Technologies, CA, USA). The library was prepared according to the standard procedures of Novogene, and its quality was assessed on the Agilent Bioanalyzer 2100 system. The clustering of the index-coded samples was performed on a cBot Cluster Generation System using TruSeq PE Cluster Kit v3-cBot-HS (Illumia) according to the manufacturer's instructions. After cluster generation, the library preparations were sequenced on an Illumina Novaseq platform, and 150 bp paired-end reads were generated. Index of the reference genome WBcel235 was built using Hisat2 v2.0.5 and paired-end clean reads were aligned to the reference genome using Hisat2 v2.0.5. Genes with a fold change of >1.5 (either up or down) and an FDR of <0.05 were considered differentially regulated genes. Gene functional enrichment was analyzed using WormCat [30]. The RNA sequencing data have been deposited in the Sequence Read Archive (SRA) with an accession number of SRP471409.

## Statistics

Data are presented as mean±SD. The GraphPad prism 10 software was used for statistical testing. Survival and life span data were analyzed using a log-rank (Mantel–Cox) test. The nucleolar occupancy of fluorescent proteins was analyzed using a Chi-square and Fisher's exact test. Other data were analyzed using an ANOVA or *t* test, as shown in the figure legends. $p < 0.05$ was considered significant. The micrographic and immunoblotting images shown are representative of at least three independent experiments that produced comparable results.

## Supporting information

**S1 Fig. RNA exosome inactivation triggers systemic activation of stress responses. (A)** Left: RNAi effects of the RNA exosome subunits on *gst-4p*::GFP expression. Scale bar = 200 μm. Right: Relative GFP intensity. One-way ANOVA with Dunnett's multiple comparisons test (*$p = 0.0129$) ($n = 30$ worms). **(B–E)** Left: Effects of *exos-8* and *exos-4.2* RNAi on the expression of MTL-1::GFP (B), *hsp-4p*::GFP (C), *hsp-6p*::GFP (D), and *hsp-16.2p*::GFP (E). These reporter constructs express GFP under the control of endogenous promoters from individual stress-response genes. Scale bar = 200 μm. Right: Relative GFP intensity. One-way ANOVA with Dunnett's multiple comparisons test (n = 20−30 worms). **(F)** Effects of *exos-8* and *exos-4.2* RNAi on the mRNA levels of stress response genes. Multiple *t* test with correction for multiple comparisons using the Holm–Sidak method (**$p = 0.0076$ for *hsp-6*, 0.003 for *gst-4*, *$p = 0.0168$ for *hsp-6*, 0.0202/0.021 for *hsp-16.2,* 0.0168/0.0589 for *mtl-1*, 0.0255 for *gst-4*) ($n = 5$ independent experiments). The numerical data presented in this figure can be found in S1 Data. Data are presented as mean±SD. *$p < 0.05$, **$p < 0.01$, ***$p < 0.001$. (TIF)

**S2 Fig. Stresses may deplete the nucleolar RNA exosome via food pumping inhibition. (A)** Effects of *exos-8* RNAi on the nucleolar localization of GFP::EXOS-1 in intestinal cells. Scale bar = 10 μm. Right: Percentage of fluorescent signals in the nucleoli. $n = 78$ and 93 cells. **(B–D)** Effects of oxidative stress (B), mitochondrial stress (C), and endoplasmic reticulum stress (D) on the nucleolar localization of GFP::EXOS-1 in intestinal cells. Scale bar = 10 μm. Right: Percentage of fluorescent signals in the nucleoli. n = 75−127 cells. **(E)** Effects of stresses on food pumping rate. One-way ANOVA ($n = 10$ worms). **(F–H)** Left: Effects of the *rsks-1* and *rict-1* mutations on the nucleolar localization of GFP::EXOS-1 (F), EXOS-8:: mCHERRY (**$p = 0.005$) (G), and mCHERRY::DIS-3 (*$p = 0.0107$, **$p = 0.0045$) (H) in intestinal cells. Scale

bar = 10 μm. Right: Percentage of fluorescent signals in the nucleoli. *n* = 78−171 cells. **(I)** Left: The nucleolar localization of GFP::EXOS-1 in intestinal cells in response to RNAi targeting *T22H9.1, snu-13, nol-56, fib-1, and mtr-4*. Scale bar = 10 μm. Right: Percentage of fluorescent signals in the nucleoli. *n* = 93−109 cells. **(J)** Effect of the *exos-10* mutation on oxidative stress resistance in well-fed and 12-h starved L4-stage nematodes. **(K)** Effects of the *rsks-1* mutation on *exos-8* RNAi-induced oxidative stress resistance. Data are presented as mean ± SD. *$p < 0.05$, **$p < 0.01$, ***$p < 0.001$. S1 Table provides all repeats and statistical analyses of the survival experiments, where Repeat 1 of each experiment was used for generating the graphs. The numerical data presented in this figure can be found in S1 Data.
(TIF)

**S3 Fig. RNA exosome inactivation does not regulate stress response genes directly or via nucleolus size.**
**(A, B)** Functional classification of *exos-8* RNAi-downregulated genes by WormCat Categories Two (A) and Three (B) analysis. **(C, D)** Functional classification of *exos-8* RNAi-upregulated genes by WormCat Categories Two (C) and Three (D) analysis. **(E–G)** Left: Effects of *exos-8* RNAi on the *mtl-1* transcriptional reporter *(mtl-1p*::GFP) (E), translational reporter *(mtl-1p*::MTL-1::GFP) (F), and ectopic expression reporter *(ges-1p*::MTL-1::GFP) (G). The *daf-16* mutation suppresses the induction of the translational reporter in *exos-8(RNAi)* worms (F). Scale bar = 200 μm. Right: Relative GFP intensity. E and G, unpaired two-tailed *t* test. F, two-way ANOVA with Turkey's multiple comparisons test (*n* = 14−30 worms). **(H)** Left: Effects of *exos-8* and *exos-4.2* RNAi on nucleolus size indicated by FIB-1::mCHERRY in intestinal cells. Scale bar = 10 μm. Right: Quantification of nucleolus size. One-way ANOVA with Dunnett's multiple comparisons test (*n* = 50 cells). **(I)** Effect of *ncl-1* mutation on *exos-8* RNAi-induced oxidative stress resistance. Data are presented as mean ± SD. ***$p < 0.001$. S1 Table provides all repeats and statistical analyses of the survival experiments, where Repeat 1 of each experiment was used for generating the graphs. The numerical data presented in this figure can be found in S1 Data.
(TIF)

**S4 Fig. RNA exosome inactivation modulates lipid metabolism. (A)** Effects of *exos-8* RNAi on the expression of lipid metabolic genes based on RNA-seq data. Fold changes were shown by colors as indicated in fig. FFA: free fatty acid; DAG: diglycerides; TAG: triglycerides; ACS: acyl-CoA synthase; ECH: enoyl CoA hydratase; ACDH: acyl-CoA dehydrogenase; CPT: carnitine palmityl *transferase*; ACO: acyl-CoA oxidase; HACD: hydroxy acyl-CoA dehydrogenase. **(B)** Left: Effects of *exos-8* and *exos-4.2* RNAi on fat accumulation measured by Nile red staining. Scale bar = 200 μm. Right: Relative Nile red intensities. One-way ANOVA with Dunnett's multiple comparisons test (*n* = 30 worms). **(C)** Left: Effect of the *exos-10* mutation on fat accumulation measured by Oil red O staining. Scale bar = 200 μm. **(D–F)** Effects of the *mxl-3* mutation (D), the nhr-49 mutation (E), and hlh-11 RNAi (F) on *exos-8* RNAi-induced lipid accumulation. Scale bar = 200 μm. **(G)** Upper: Effects of *exos-8* and *exos-4.2* RNAi on the protein levels of SBP-1::GFP. Lower: Quantification of western blot. One-way ANOVA with Dunnett's multiple comparisons test (**$p = 0.0017$) (*n* = 3 experiments). **(H)** Effect of the *exos-10* mutation on lipid accumulation in well-fed and starved animals. Data are presented as mean ± SD. *$p < 0.05$, ***$p < 0.001$. The numerical data presented in this figure can be found in S1 Data.
(TIF)

**S5 Fig. The 1CM pathways upstream of SAMS-1 are not involved in the cellular responses to RNA exosome inactivation. (A)** RNA-seq data reveal regulation of nucleotide metabolic enzymes by *exos-8 RNAi*. False Discovery Rate (FDR) calculated by the Benjamini and Hochberg's approach (*n* = 4 experimental groups). **(B)** Effect of the *sams-1* mutation on *exos-10* mutation-induced lipid accumulation. Scale bar = 200 μm. **(C)** Effect of the *sams-1* mutation on *exos-10* mutation-induced mitochondrial stress resistance. **(D–F)** Supplementations of serine (D), glycine (E), or methionine (F) have no effects on *exos-8* RNAi-induced lipid accumulation. Scale bar = 200 μm. **(G)** Knockdowns of the 1CM genes have little or no effects on *exos-8* RNAi-induced lipid accumulation. Scale bar = 200 μm. **(H)** Knockdown of *pmt-2* inhibits *exos-8* RNAi-induced lipid accumulation. Scale bar = 200 μm. Data are presented as mean ± SD. **$p < 0.01$, ***$p < 0.001$. S1 Table

provides all repeats and statistical analyses of the survival experiments, where Repeat 1 of each experiment was used for generating the graphs. The numerical data presented in this figure can be found in S1 Data.
(TIF)

**S6 Fig. Translation inhibition preserves fat and improves stress resistance in response to RNA exosome inactivation. (A)** Effects of PC supplementation on lipid accumulation in *sams-1* mutants. Scale bar = 200 μm. **(B)** Effects of PC supplementation on *exos-8* RNAi-induced lipid accumulation. Scale bar = 200 μm. **(C)** RNA-seq data reveal translation initiation factors that were downregulated by *exos-8* RNAi. **(D)** Left: Effect of the *exos-10* mutation on translation. Right: Quantification of translation. Paired two-tailed *t* test (*$p$ = 0.0116) ($n$ = 4 experiments). **(E, F)** Effects of *eif-2β* RNAi on *exos-8* RNAi-induced lipid accumulation (D) and oxidative stress resistance (E). Scale bar = 200 μm. **(G)** Effect of the translation inhibitor cycloheximide (CHX) on *exos-10* mutation-induced oxidative stress resistance. **(H)** Left: Starvation suppresses cellular translation. Right: Quantification of translation. *t* test ($n$ = 3 experiments). Data are presented as mean ± SD. *$p < 0.05$, ***$p < 0.001$. S1 Table provides all repeats and statistical analyses of the survival experiments, where Repeat 1 of each experiment was used for generating the graphs. The numerical data presented in this figure can be found in S1 Data.
(TIF)

**S7 Fig. The life span-extending effect of RNA exosome inactivation is specific to nutrient deprivation. (A, B)** Effects of post-developmental *exos-8* RNAi on oxidative stress resistance (A) and mitochondrial stress resistance (B). **(C)** Effects of *exos-8* RNAi on oxidative stress resistance in *daf-2* mutants. **(D, E)** Effects of *exos-8* RNAi from birth (D) and after development (E) on the life span of *daf-2* mutants. S1 and S2 Tables provide all repeats and statistical analyses of the survival experiments, where Repeat 1 of each experiment was used for generating the graphs.
(TIF)

**S1 Table. Survival data. Results shown are representative of at least two independent experiments. Repeats 1 were graphed in figures.**
(DOCX)

**S2 Table. Lifespan data. Results shown are representative of at least two independent experiments. Repeats 1 were graphed in figures.**
(DOCX)

**S3 Table. The knockdown efficacy of RNAi.**
(DOCX)

**S4 Table. Primers for cRT-PCR and qPCR.**
(DOCX)

**S5 Table. Genes differentially regulated by *exos-8* RNAi identified via RNA-seq analysis.**
(XLSX)

**S1 Data. The underlying numerical data in the manuscript.**
(XLSX)

**S1 Raw Images. Original blots and gels in this manuscript.**
(PDF)

## Acknowledgments

We thank CGC and Dr. Bin Liang for providing the strains. We thank Junling Liu, Feng Li, and Xiangyang Chen for technical assistance. We also thank Analytic and Testing Center of Chongqing University for the use of facility and technical support.

 

## Author contributions

**Conceptualization:** Shanshan Pang, Haiqing Tang.

**Data curation:** Xi Feng, Haiqing Tang.

**Formal analysis:** Xi Feng, Xiaoman Wang, Shanshan Pang, Haiqing Tang.

**Funding acquisition:** Shanshan Pang, Haiqing Tang.

**Investigation:** Xi Feng, Xiaoman Wang, Haiqing Tang.

**Methodology:** Xi Feng, Shouhong Guang, Shanshan Pang, Haiqing Tang.

**Project administration:** Shanshan Pang, Haiqing Tang.

**Resources:** Shouhong Guang, Shanshan Pang, Haiqing Tang.

**Supervision:** Shanshan Pang, Haiqing Tang.

**Writing – original draft:** Shanshan Pang, Haiqing Tang.

**Writing – review & editing:** Xi Feng, Shanshan Pang, Haiqing Tang.

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
