## [Editor Report · Decision Letter 0]

8 Jan 2025

Dear Dr Tang, 

Thank you for submitting the revised version of your manuscript entitled "Inhibition of the nucleolar RNA exosome facilitates starvation adaptation" for consideration as a Research Article by PLOS Biology, and please accept my apologies for the delay due to the holidays.

Your revision has now been evaluated by the PLOS Biology editorial staff as well as the original academic editor and I am writing to let you know that we would like to send it back to the original reviewers. However, as this is a new submission, we will need you to complete again your submission by providing the metadata.

To this end, please login to Editorial Manager where you will find the paper in the 'Submissions Needing Revisions' folder on your homepage. Please click 'Revise Submission' from the Action Links and complete all additional questions in the submission questionnaire.

Once your full submission is complete, your paper will undergo a series of checks in preparation for peer review. After your manuscript has passed the checks it will be sent out for review. To provide the metadata for your submission, please Login to Editorial Manager (https://www.editorialmanager.com/pbiology) within two working days, i.e. by Jan 10 2025 11:59PM.

Kind regards,

Ines

--

Ines Alvarez-Garcia, PhD

Senior Editor

PLOS Biology

---

## [Decision Letter · Decision Letter 1]

21 Mar 2025

Dear Dr Tang,

Thank you for your patience while we considered your revised manuscript entitled "Inhibition of the nucleolar RNA exosome facilitates starvation adaptation" for publication as a Research Article at PLOS Biology. This revised version of your manuscript has been evaluated by the PLOS Biology editors, the Academic Editor and one of the original reviewers.

Based on the review (attached below), we are likely to accept this manuscript for publication, provided you satisfactorily address the remaining points raised by the reviewer. Please also make sure to address the data and other policy-related requests stated below my signature.

In addition, we would like you to consider a suggestion to improve the title:

"Inhibition of the nucleolar RNA exosome facilitates adaptation to starvation"

We expect to receive your revised manuscript within two weeks. 

*Published Peer Review History*

*Press*

Sincerely,

Ines

--

Ines Alvarez-Garcia, PhD

Senior Editor

PLOS Biology

Fig. 1A-E; Fig. 2A-D, H-J; Fig. 3A-L; Fig. 4B-G; Fig. 5A-H; Fig. 6A-G; Fig. S1B-K; Fig. S2A-K; Fig. S3A-I; Fig. S4B, G; Fig. S5A, C; Fig. S6D, F-H and Fig. S7A-E

CODE POLICY

We require the original, uncropped and minimally adjusted images supporting all blot and gel results reported in an article's figures or Supporting Information files. We will require these files before a manuscript can be accepted so please prepare and upload them now. We would need the original gels shown in the following figures: Fig. 2F, G; Fig. 5A-D; Fig. 6F and Fig. S6D, H

Please carefully read our guidelines for how to prepare and upload this data: https://journals.plos.org/plosbiology/s/figures#loc-blot-and-gel-reporting-requirements

Reviewers' comments

Rev. 1:

In the revised version of their article, Feng et al have addressed my concerns significantly though not completely. I am happy to support publication of the manuscript in PLoS Biology. I have a few comments/suggestions below that I recommend be applied to the article before publication.

1. It will help if the figure legends of the survival graphs provide more information in terms of mean survival value, SEM, P values. It is great that the Suppl. Tables list these data from the different trials, but adding the specific values in the legend (or indicating which trial # is being reported in the figure) will improve the reading experience.

2. Similarly, for experiments such as ORO staining and qPCRs, it is important to indicate how many trials/biological replicates were performed and what the no. of animals in each was. And, if the quantification is derived from combining all or select replicates.

3. I strongly recommend moving the exos-10 mutant data (Fig. S1) to the main manuscript along with the schematic of the exosome.

4. It is not clear how Fig. 3A, B is/are different from Fig. 3C, D?

5. Where are these exosome genes expressed? Were the reporter constructs used in Fig. 1 etc,. driven under endogenous promoters or in specific tissues? Which tissues was the imaging done on- one or more? This information is either not included or poorly accessible. Please include this. It will help if there is also some discussion about the spatial aspects of this regulatory phenomenon.

---

## [Editor Report · Decision Letter 2]

30 Apr 2025

Dear Dr Tang,

Thank you for the submission of your revised Research Article entitled "Inhibition of the nucleolar RNA exosome facilitates adaptation to starvation" for publication in PLOS Biology. On behalf of my colleagues and the Academic Editor, Alex Gould, I am delighted to let you know that we can in principle accept your manuscript for publication, provided you address any remaining formatting and reporting issues. These will be detailed in an email you should receive within 2-3 business days from our colleagues in the journal operations team; no action is required from you until then. Please note that we will not be able to formally accept your manuscript and schedule it for publication until you have completed any requested changes.

PRESS

Sincerely, 

Ines

--

Ines Alvarez-Garcia, PhD

Senior Editor

PLOS Biology
